# Neuropilin-2 and Its Transcript Variants Correlate with Clinical Outcome in Bladder Cancer

**DOI:** 10.3390/genes12040550

**Published:** 2021-04-09

**Authors:** Sarah Förster, Maryam Givehchi, Katja Nitschke, Thomas Mayr, Kerstin Kilian, Samikshan Dutta, Kaustubh Datta, Philipp Nuhn, Zoran Popovic, Michael H. Muders, Philipp Erben

**Affiliations:** 1Rudolf Becker Laboratory for Prostate Cancer Research, Institute of Pathology, University of Bonn Medical Center, 53127 Bonn, Germany; Sarah.Foerster@ukbonn.de (S.F.); thomas.mayr@ukbonn.de (T.M.); 2Department of Urology and Urosurgery, Medical Faculty Mannheim, University of Heidelberg, 68167 Mannheim, Germany; m.givehchi@hotmail.de (M.G.); Katja.Nitschke@medma.uni-heidelberg.de (K.N.); kerstin.kilian@pkilian.de (K.K.); Philipp.Nuhn@medma.uni-heidelberg.de (P.N.); 3Department of Biochemistry, University of Nebraska Medical Center, Omaha, NE 68198-5870, USA; samikshan.dutta@unmc.edu (S.D.); kaustubh.datta@unmc.edu (K.D.); 4Institute of Pathology, Medical Faculty Mannheim, University of Heidelberg, 68167 Mannheim, Germany; zoran.popovic@umm.de; 5Bridge (Bladder Cancer Research Initiative for Drug Targets Germany) Consortium e.V., 68135 Mannheim, Germany

**Keywords:** neuropilin-2 (NRP2), neuropilin-2 transcript variants, NRP2a, bladder cancer, muscle-invasive bladder cancer (MIBC), platelet-derived growth factor (PDGF)

## Abstract

Urothelial bladder cancer ranks among the 10 most frequently diagnosed cancers worldwide. In our previous study, the transmembrane protein neuropilin-2 (NRP2) emerged as a predictive marker in patients with bladder cancer. NRP2 consists of several splice variants; the most abundant of these, NRP2a and NRP2b, are reported to have different biological functions in lung cancer progression. For other cancer types, there are no published data on the role of these transcript variants in cancer progression and the clinical outcome. Here, we correlate *NRP2* and its two most abundant transcript variants, *NRP2A* and *NRP2B*, with the clinical outcome using available genomic data with subsequent validation in our own cohort of patients with muscle-invasive bladder cancer. In addition to *NRP2*, *NRP1* and the NRP ligands *PDGFC* and *PDGFD* were studied. Only *NRP2A* emerged as an independent prognostic marker for shorter cancer-specific survival in muscle-invasive bladder cancer in our cohort of 102 patients who underwent radical cystectomy between 2008 and 2014 with a median follow-up time of 82 months. Additionally, we demonstrate that high messenger expression of *NRP2*, *NRP1*, *PDGFC* and *PDGFD* associates with a more aggressive disease (i.e., a high T stage, positive lymph node status and reduced survival).

## 1. Introduction

Urothelial bladder cancer (UBC) ranks among the 10 most frequently diagnosed cancers worldwide [1] and can be classified into non-muscle-invasive bladder cancer (NMIBC) and muscle-invasive bladder cancer (MIBC; approx. 20% of newly diagnosed cases). MIBC is more aggressive and is associated with a worse prognosis, i.e., five year survival rates of 60% for patients with localized disease but <10% for metastatic disease [2]. Despite aggressive therapy regimens, i.e., radical cystectomy (RC) and (neo)adjuvant platinum-based chemotherapy, MIBC often progresses to metastatic disease. In the metastatic setting curative therapy options are limited [3].

MIBC is a heterogeneous disease that shows high overall mutation rates. Based on transcriptome profiling/mRNA expression, MIBC can be divided into five molecular subtypes: basal-squamous, luminal-papillary, luminal-infiltrated, luminal and neuronal subtypes [4]. These molecular subtypes are of clinical significance as they may be used to stratify patients for prognosis and response to chemotherapy or immunotherapy [2,4]. Novel markers may help to identify patients with an increased risk for cancer progression but might also be used as predictors for therapy success. Targeting these markers could lead to novel therapeutic avenues for patients with advanced disease. In recent years, neuropilin (NRP)-2 has emerged as one such target.

The NRP family consists of the two structurally homologous transmembrane proteins NRP1 and NRP2, located on chromosomes 10p12 and 2q34, respectively [5]. NRPs are co-receptors for selected members of the family of vascular endothelial growth factors (VEGFs) as well as several class 3 semaphorins and have been implicated in cancer angiogenesis and lymphangiogenesis [5,6]. In fact, NRP2 is frequently overexpressed in tumors and is associated with a poor prognosis in various cancers [7,8,9,10,11]. Using a cohort of patients with UBC treated with a transurethral resection of the bladder tumor (TURBT) and adjuvant radio(chemo)therapy, we have previously shown that NRP2 protein expression is a predictive marker for overall survival (OS) as well as cancer-specific survival (CSS) in NMIBC and MIBC [12]. To our knowledge, no data are available on the prognostic value of NRP2 in treatment-naïve MIBC patients treated with an RC. In addition, alternative splicing of the *NRP2* gene gives rise to several NRP2 transcripts/isoforms. To date, insufficient data are available on *NRP2* transcript-specific associations with histopathological parameters and cancer prognosis.

In addition to semaphorins and VEGFs, other heparin-binding growth factors such as platelet-derived growth factors (PDGFs) have also been described as binding to NRPs [13,14] and several reports have indicated the involvement of NRPs in PDGF signaling [15,16,17]. The PDGF family consists of four members (PDGF-A–D) that are secreted as homodimers or heterodimers (AA, AB, BB, CC or DD) and bind to and signal via PDGF receptors. Like NRPs, PDGFs and PDGF receptors are abundantly expressed by cells in the tumor microenvironment (e.g., endothelial cells, vascular smooth muscle cells and tumor-associated macrophages) but also by tumor cells (reviewed in [18]). Due to their more recent discovery, the roles of PDGF-C and PDGF-D in cancer are less well studied.

The aim of this study was to investigate the clinical relevance of *NRP2* and its transcript variants in MIBC using data from the “The Cancer Genome Atlas” (TCGA) bladder cancer (BLCA) cohort. We further validated these results in a retrospective single center cohort. Associations of *NRP2*, *NRP1* and several NRP ligands (i.e., *PDGFC* and *PDGFD*) that have been implicated in cancer progression were analyzed.

## 2. Materials and Methods

### 2.1. Cohort and Patient Samples 

A cohort of 102 patients (men: *n* = 73, 71.57%, median age: 71.2 years; range: 46.0–87.4 years; female: *n* = 29, 28.43%, median age: 67.9 years; range: 49.2–86.5) who underwent an RC at the Department of Urology and Urosurgery of the University of Mannheim Medical Center between 2008 and 2014 (Mannheim cohort) was used for the validation of TCGA data. The exclusion criteria for the individual analyses are shown in Figure 1. The pathological stage of the formalin-fixed paraffin-embedded (FFPE) tumor tissue samples was evaluated according to the 2017 TNM classification of the Union for International Cancer Control (UICC) and grading was performed according to the 2017 WHO/ISUP classification [19,20]. The study was conducted according to the guidelines of the Declaration of Helsinki and approved by the Ethics Committee of the University of Heidelberg (ethics approval 2015-549N-MA; date of approval 25 May 2015; date of approval of first amendment 12 July 2018; date of approval of second amendment 10 December 2020). All patients gave informed consent for participation.

### 2.2. RNA Extraction, cDNA Synthesis and qRT-PCR Analyses of Patient Samples

Tumor-bearing FFPE urinary bladder tissue specimens were stained with hematoxylin and eosin and reviewed by a board-certified surgical uropathologist (PZ). RNA was extracted using the magnetic bead-based XTRAKT FFPE kit (Stratifyer, Cologne, Germany) according to the instruction of manufacturer. Finally, the RNA was eluted in 100 μL of elution buffer and stored at −80 °C. cDNA synthesis with a pool of sequence-specific reverse PCR primers (reference genes *CALM2* and target genes *NRP1*, *NRP2*, *NRP2A*, *NRP2B*, *PDGFC*, *PDGFD*) was performed. Superscript III (Thermo Fisher Scientific, Waltham, MA, USA) was used as reverse transcriptase at 55 °C for 120 min, followed by an enzyme inactivation step at 70 °C for 15 min. cDNA was stored at −20 °C or directly used for qPCR. A total of 40 cycles of amplification with 3 s of 95 °C and 30 s of 60 °C were performed on a StepOnePlus qRT-PCR cycler (Applied Biosystems, Waltham, MA, USA). The gene expression was normalized to the reference gene *CALM2* and determined using the 40-(ΔCt) method [21,22]. Appendix A shows all primers and probes used in this study. Primers were specifically designed to generate short (100–150 bp) variant specific amplification products from FFPE material.

### 2.3. TCGA Cohort and Statistics

A TCGA sequencing dataset was obtained from USC Xena Browser [23] and cBioPortal [24,25] (http://www.cbioportal.org; last accessed on 20.01.2021). Data on NRP2 splice variant expressions were downloaded from the TCGA Splice Variant database [26] (TSVdb). mRNA was extracted from fresh frozen samples, prepared into libraries and sequenced by Illumina HiSeq as described [4]. The TCGA dataset was log2 transformed and contained RNA sequencing data of 413 patients with MIBC as well as clinicopathological and follow-up data. Metastatic samples, patients with a prior neoadjuvant treatment, unknown T stage and T stage < 2 as well as patients with missing gene expression data were excluded (Figure 2). Finally, the expression and clinical data of 360 patients with MIBC were reanalyzed as described below. 

### 2.4. Statistics

The analyses of both cohorts were performed using the methods described below. Statistical analyses were performed using SAS Jmp 14 and GraphPad Prism 5 (GraphPad Software, La Jolla, CA, USA). Mann–Whitney U tests were used to compare the data characterized by a non-normal distribution. A Spearman test was used to analyze the correlation between the expression of different genes. The cut-off values for high and low gene expressions were determined by a partition test with each group representing at least 20% of the total cohort [27,28]. The Kaplan–Meier method and the log-rank test were used for survival analyses and for the Cox regression analysis univariable and multivariable hazard ratios (HR) were used. A multivariable analysis was performed for variables with *p* ≤ 0.1. All tests were performed two-sided and *p* values of <0.05 were considered statistically significant.

## 3. Results

### 3.1. Patient Cohorts and Clinicopathological Characteristics

The described TGCA dataset of histologically confirmed and neoadjuvant-naïve MIBC (*n* = 360) was used for our analyses. The demographic and clinicopathological characteristics of the final TCGA dataset are listed in Table 1. While the demographic and clinicopathological distribution of the Mannheim cohort (Table 1) was similar to the TCGA dataset, the TCGA cohort was larger than the Mannheim cohort (360 vs. 102 patients) but the Mannheim cohort had significantly longer follow-up times (TCGA: 29.8 months vs. Mannheim: 82 months). 

### 3.2. Analysis of NRP2 Isoform Expression 

TSVdb data on NRP2 splice variants showed that six NRP2 variants, namely NRP2A22, NRP2A17, NRP2A0, NRP2B5, NRP2B0 and S9NRP2 (or NRP2 transcript variants 1–6), are generated from the *NRP2* gene and expressed in a cohort with MIBC (Figure 3). All samples (*n* = 360) expressed at least one NRP2A transcript and 99% (357 of 360) expressed at least one NRP2B transcript and 67% (243 of 360) expressed NRP2S9. For further analysis, NRP2A22, NRP2A17 and NRP2A0 as well as NRP2B5 and NRP2B0 were combined into NRP2A and NRP2B, respectively, and data were log2 transformed. NRP2A expression was always higher than NRP2B expression in the TCGA cohort (Appendix A). A positive ratio of NRP2A to NRP2B expression was also observed in most cases (93.1%) in the Mannheim cohort (Appendix A).

### 3.3. Marker Expression

The co-expression of selected markers is shown in Appendix A. In the TCGA cohort, NRP2 positively correlated with NRP1, PDGFC and PDGFD. The correlation between NRP2 and PGDFC was strongest (ρ = 0.74, *p* < 0.0001) while NRP2 and PDGFD were only weakly correlated (ρ = 0.12, *p* = 0.024). When analyzing NRP2 in more detail, NRP2A and NRP2B variants were similarly correlated with NRP1 and PDGFC (ρ = 0.65 and ρ = 0.64 or ρ = 0.74 and ρ = 0.68, respectively, *p* < 0.0001 for all). However, while NRP2A showed a weak but significant correlation with PDGFD, the correlation of NRP2B and PDGFD did not reach statistical significance (ρ = 0.12; *p* = 0.025 and ρ = 0.05; *p* = 0.32, respectively). In the Mannheim cohort, the marker expression was generally very similar. Of note, a stronger and significant correlation of PDGFD with all markers was observed in the validation cohort.

### 3.4. Associations Between Marker Expression and Clinicopathological Characteristics

The bivariate analysis of gene expression and clinicopathological parameters in the TGCA cohort is shown in Table 2. In the TCGA cohort, NRP2 and its variants significantly correlated with a higher tumor stage (T3/4 vs. T2), positive lymph node metastasis and recurrence. NRP1 significantly associated with the tumor stage. PDGFC significantly associated with the tumor stage, lymph node metastasis and recurrence. Only PDGFD significantly associated with lymphovascular invasion (LVI). Similar trends could be observed for all other genes yet these associations did not reach a statistical significance. Lastly, the association of gene expression with molecular subtype was investigated. Based on the molecular signatures, the TCGA samples were grouped into basal-squamous (*n* = 132), luminal, luminal-infiltrated, luminal-papillary and neuronal (*n* = 25, 69, 118, 16, respectively). The molecular subtype complexity was reduced by dichotomizing into basal vs. not basal type. For all investigated genes, the gene expression significantly associated with the molecular subtype in the TCGA cohort. With the exception of *PDGFD*, a higher gene expression was always observed in the basal subtype.

In the Mannheim cohort, the NRP1, PDGFC and PDGFD gene expressions were significantly associated with LVI. NRP2 and NRP2A but not NRP2B showed similar trends albeit not reaching statistical significance (Table 3).

### 3.5. Survival Analysis and Univariable and Multivariable Analysis for Overall Survival (OS)

Kaplan–Meier curves of OS and their association with gene expression are depicted in Figure 4. For all investigated markers, a high gene expression was associated with a significantly shorter OS. In the univariable analysis, a high age, a high T stage, locoregional lymph node involvement, the presence of LVI and a high expression of all investigated genes significantly associated with a reduced OS. Only age and PDGFD were prognostic markers for a reduced OS (Table 4). Similar results were obtained for disease-free survival (DFS) (Appendix A). 

Subgroup analyses were also performed after stratification for T stage (T2 and T3/4) and locoregional lymph node metastasis (Nneg and Npos) as well as a high expression of NRP2 or its variants. The results of the Kaplan–Meier survival analysis and log-rank test are listed in Table 5. In general, survival analyses after stratification for T or N stage showed similar results with slightly elevated *p* values when compared with the entire cohort. Of note, NRP1 expression was only significantly associated with the OS for patients with T2 but not T3/4 of disease (*p* = 0.013 vs. *p* = 0.18). In contrast, the association of PDGFD with the OS was significantly stronger in the T3/4 subgroup (*p* < 0.0001 vs. *p* = 0.024 for T2). After stratification for Nneg patients, PDGFD lost its significant association with the OS. However, when stratifying for Npos disease, PDGFC and PDGFD were the only variables significantly associated with the OS (*p* = 0.005 and *p* = < 0.0001, respectively). Lastly, marker combinations and their implications for the OS were investigated using a Kaplan–Meier analysis (Figure 5). Patients with a high NRP2 expression combined with a high NRP1, PDGFC or PDGFD expression showed drastically reduced OS. These effects were less pronounced when investigating the other marker combinations.

### 3.6. Survival Analysis and Univariate and Multivariable Analysis for Cancer-Specific Survival (CSS)

While the TCGA data showed significant associations of gene expression with the OS and DFS, we were mainly interested in cancer-specific survival (CSS). These data were not available for the TCGA cohort. Hence, CSS was investigated in the Mannheim cohort. In the Kaplan–Meier analysis, NRP2A was significantly associated with CSS while this was not true for NRP2 and NRP2B (Figure 6). With regard to NRP1, data showed a trend towards a lower CSS for patients with a high expression but this association did not reach statistical significance (*p* = 0.09). In the univariable analysis, only the T stage and NRP2A expression were significantly associated with a reduced CSS (*p* = 0.003 and *p* = 0.024, respectively). A multivariable analysis revealed a high T stage and a high NRP2A expression as independent prognostic markers for a shorter CSS (Table 6). Due to the smaller sample size in the Mannheim cohort, only T3/4 stage and Nneg groups were investigated separately, resulting in similar results as described for the whole cohort (Table 7).

## 4. Discussion

MIBC often progresses to metastatic disease yet curative treatment options are currently limited in the metastatic setting. Novel predictive and prognostic markers and therapy targets are urgently needed for treatment stratification. In addition, these markers might emerge as targets for novel treatment avenues. Recent studies have demonstrated that the important role of NRP2 in cancer progression and metastases might qualify this transmembrane receptor as a potential therapeutic target [6,30]. Our group has previously demonstrated that NRP2 protein was a predictive factor for the outcome in a special patient cohort with several comorbidities that suffered from T2–4 or high risk T1 BLCA and that were treated with TURBT and adjuvant radiochemotherapy. High NRP2 protein levels were associated with reduced OS and CSS. A multivariate Cox regression analysis also revealed NRP2 as an independent prognostic factor for OS in this cohort [12]. The role of NRP2 in the therapy response was corroborated in vitro where we showed that NRP2 downregulation sensitized BLCA cells to radiochemotherapy [31]. However, the prognostic value of NRP2 may be limited to MIBC. In fact, NRP2 was not associated with tumor grade and stage and failed to predict recurrence/progression in an NMIBC cohort consisting of cases with superficial (pTa) and mucosa-invasive (pT1) tumors [32]. Similar results were obtained in a cohort with mixed tumor stages (Tis/Ta–T4). While *NRP2* gene expression as well as protein levels could be employed to separate early-stage and invasive UBC lesions, across the entire spectrum of bladder cancer progression from superficial to invasive lesions NRP2 was not associated with OS [33]. Due to the significant risk of progression, a worse prognosis and more limited treatment options, we focused our current study on MIBC. Before the discussion of the results obtained in our current study, it should be pointed out that the TCGA cohort analyzed fresh frozen material while the Mannheim cohort consisted of FFPE material. While fixation is known to affect the sample, FFPE material has the great advantage of being routinely collected and stored in the clinic usually making it more easily accessible. In the TCGA cohort, the *NRP2* gene expression was significantly associated with an increased tumor stage as well as locoregional lymph node metastasis. While the total *NRP2* messenger level was significantly associated with OS and DFS, we could not show an association of total *NRP2* expression with CSS in the Mannheim cohort. These discrepancies may arise due to differences in study cohorts (sample acquisition by RC vs. TURBT; treatment-naïve patients vs. radiochemotherapy; sample size; sample type (fresh frozen vs. FFPE); follow-up time). Furthermore, our current study investigated *NRP2* gene expression while Keck et al. investigated NRP2 protein levels [12]. In light of our reports indicating post-transcriptional regulation that may influence NRP2 protein levels [31], a direct comparison of immunohistochemical NRP2 staining with mRNA expression should be performed in the future to clarify this issue. However, our current study was particularly focused on messenger expression as we were interested in the potential involvement of NRP2 isoforms/transcript variants in UBC but no isoform-specific antibodies are commercially available yet. 

The alternative splicing of the *NRP2* gene gives rise to several NRP2 transcripts/isoforms. The transmembrane proteins NRP2a and NRP2b have identical extracellular N-terminal domains but differ significantly in their juxtamembrane, transmembrane and cytoplasmic domains. Indeed, a comparison of these domains revealed that NRP2a and NRP1 are much closer in sequence than NRP2a and NRP2b (44% vs. 11% sequence identity) [29]. Making use of publicly available splice variant data from the TSVdb, we showed that bladder cancer specimens expressed both *NRP2A* and *NRP2B* as well as *S9NRP2* transcripts. Both *NRP2A* and *NRP2B* can be alternatively spliced and give rise to different splice variants, named after the number of amino acid insertions in the C-terminal domain (i.e., A0, A17 and A22 or B0 and B5) [29,34]. For *NRP2A*, *A22* was the most abundant and highly expressed variant in BLCA followed by *NRP2A17* and *A0*. Regarding *NRP2B* splice variants, *NRP2B0* was more abundant than *NRP2B5* in the TCGA cohort. To the best of our knowledge, the expression profiles of *NRP2* splice variants in human tissue have been investigated in very few studies. Using a Northern blot analysis, Rossignol et al. investigated the brain, heart, kidney, lung, liver, placenta and trachea and also found *NRP2A17* to be more abundant than *NRP2A22* while *NRP2A0* was not detected at all. In the investigated tissues, *NRP2B0* was also more abundant than *NRP2B5* [29]. Whether the observed differences regarding the *NRP2A0* variant were due to organ/tissue-specific differences or could be attributed to a differential expression in cancer vs. normal tissue remains to be investigated. Another interesting finding of this Northern blot analysis was that the ratio of *NRP2A* to *NRP2B* may be tissue-specific. For example, in lung and liver tissues, *NRP2A* expression was markedly higher than *NRP2B* expression, while in heart and skeletal muscle, *NRP2B* was more abundant than *NRP2A.* Another study investigating *NRP2* transcripts in human lipopolysaccharide-stimulated dendritic cells indicated that *NRP2A*/*NRP2B* ratios may also be patient-specific [35]. Intriguingly, in the 360 patients investigated from the TCGA cohort, the *NRP2A* expression was always higher than the *NRP2B* expression. This was also true for most patients in the Mannheim cohort as well as in several bladder cancer cell lines (personal communication [36]). In the TCGA MIBC cohort, the *NRP2A* and *NRP2B* transcripts showed very similar results with regard to the OS when compared with the total *NRP2* gene expression. *NRP2* and its splice variants were significantly associated with OS and DFS. These results could not be validated in the Mannheim cohort. In the Mannheim cohort, only *NRP2A* was significantly associated with CSS. Indeed, a high *NRP2A* was associated with a reduced CSS. These discrepancies may arise due to different sample collections and preparations (e.g., fresh frozen vs. FFPE material and RNASeq vs. qRT-PCR) in the TCGA and Mannheim cohorts. Investigations using another independent cohort could clarify these issues but unfortunately we do not have access to another dataset providing information on *NRP2* transcripts at the moment. To date, only a few other studies have focused on the biological functions of NRP2 isoforms. Indeed, Gemmil et al. provided the first study investigating differential NRP2 isoform implications in vivo. In lung cancer, a high NRP2b expression was associated with a high tumor stage. Intriguingly, no association was found between total NRP2 protein levels and the tumor stage. Furthermore, a poor outcome, i.e., progression-free survival, was significantly correlated with NRP2b but not with NRP2 total protein expression in these patients. A novel NRP2b-specific antibody raised against the cytoplasmic domain of NRP2b was first developed and described in the aforementioned study but is not yet commercially available [37]. In contrast, immunohistochemical analyses performed by us and others usually employ NRP2 antibodies that bind to the N-terminus and thus detect both NRP2a and NRP2b isoforms as well as the soluble s9NRP2 [12,33]. Our results seemed to contradict Gemmill´s finding. However, a major difference between these studies is that we investigated *NRP2* and its isoforms/transcripts at the mRNA level while Gemmill et al. developed antibodies to investigate NRP2 and NRP2b isoforms at the protein level. To date, no NRP2a-specific antibody has been described. Furthermore, mRNA expression and protein levels of NRP2 may not be correlated. In bladder cancer cell lines, TGFβ1 treatment significantly elevated *NRP2* mRNA by five-fold while only a minor increase of the NRP2 protein level was observed in our hands [31]. Furthermore, NRP2 isoforms may have distinct turnover rates as indicated in lung cancer cell lines where the half-life of NRP2b was approximately twice as long as that for NRP2a [37]. It is also feasible that NRP2 isoform levels vary between bladder and lung tissue or that NRP2 isoforms are differentially involved in bladder cancer vs. lung cancer. Further studies will be needed to clarify this issue.

NRPs can form homodimers and heterodimers and interact with similar ligands. While our data showed a positive correlation between NRP2 and NRP1 expression, in vitro results regarding NRP interaction/regulation are inconsistent [38,39]. Similar to NRP2, NRP1 is expressed on various types of tumor cells and its expression correlates with tumor progression or a poorer prognosis in several cancers such as prostate, breast, non-small-cell lung carcinoma (NSCLC) and glioma [6,40,41]. In the TCGA cohort, a high *NRP1* gene expression was associated with a higher T stage and a similar, yet not statistically significant, association was observed in the Mannheim cohort. Furthermore, a high *NRP1* was associated with a reduced OS and showed a trend towards a reduced CSS (*p* = 0.09). Similar findings have been observed by others. Cheng et al. recently showed that a high NRP1 protein level was associated with the tumor stage and a reduced OS in BLCA [40].

To the best of our knowledge, this is also one of the first reports investigating the clinical implications of the NRP ligands PDGFC and PDGFD in BLCA. Several cell culture studies indicated NRPs in PDGF signaling. In smooth muscle cells, PDGF-BB upregulates *NRP1* mRNA levels and vice versa; *NRP1* as well as *NRP2* knockdown reduced PDGF-AA and/or PDGF-BB-induced PDGF receptor phosphorylation [15,16,17]. Here, we also observed a positive correlation between *NRP2* and its transcripts and *PDGFC* (and in the Mannheim cohort also *PDGFD*). All transmembrane NRP2 proteins have the same extracellular ligand-binding domains and could thus interact with the same ligands. To date, it is unclear how PDGF and NRP expressions are linked mechanistically. In our current study, *PDGFD* was significantly associated with LVI and turned out to be an independent prognostic marker for OS and DFS but not for CSS. In the TCGA cohort, a high *PDGFC* was observed in a higher T stage and lymph node positive tumors. These findings were in line with reports in other tumor entities. Bartoschek and Pietras recently explored the prognostic value of *PDGFs* and *PDGFRs* in TGCA data from 16 different tumor types. When investigating gene signatures based on the correlation with the respective PDGF family members, a high *PDGFC* gene signature showed a trend towards a reduced OS in BLCA [41]. The PDGFD protein level was determined to be an independent prognostic factor in gastric cancer while PDGFD protein levels failed to predict a recurrence in prostate cancer [42,43]. PDGFs were also found to be prognostic factors in patients with NSCLC. Intriguingly, the prognostic value of PDGFs was dependent on the tumor vs. the stromal expression of the analyzed proteins, i.e., stromal PDGFD could predict CSS; in contrast, the tumor but not the stromal PDGFC could predict CSS [44]. High PDGFD protein levels also positively correlated with a higher T stage and positive lymph node status in colorectal cancer [45] while a high PDGFC protein level has been shown to correlate with positive lymph node status in breast cancer [46]. 

Due to the lack of commercially available NRP2 isoform-specific antibodies, we did not perform any immunohistochemical analyses in this study. Nonetheless, staining for the other investigated proteins in a subsequent study will provide additional information. On the one hand, protein/marker localization and potential differential expression could be studied in more detail. The human protein atlas (HPA) provides some information on the protein expression of NRP2 in bladder cancer tissue. According to the data in the HPA, a subgroup of cancer tissues shows no expression of NRP2 protein while most of the urothelial cancer tissues express high levels of NRP2. NRP1 can also be detected in high amounts in urothelial cancer cells. In general, due to stromal and immune cell infiltration, tumor samples contain a mixture of cells with potentially distinct expression levels of specific genes. Determining tumor purity can provide important additional information. Hence, pathologists routinely determine tumor purity based on hematoxylin and eosin stained slides and combine this information with immunohistochemical stainings. According to the HPA, most of the staining was localized in the tumor cells. In addition, tumor purity can be investigated using ABSOLUTE or ESTIMATE methods based on genomic or transcriptomic information [47,48]. Indeed, in the TCGA cohort of muscle-invasive tumors, the tumor purity (based on ABSOLUTE) significantly decreased while the stromal score and ESTIMATE score significantly increased in the more aggressive disease, i.e., T stage (see Appendix A). On the other hand, a direct comparison between protein levels and mRNA expression will provide an important insight as to whether protein and mRNA levels of the investigated marker are indeed correlated. While this comparison was beyond the scope of our current project, future research would greatly benefit from the combination of immunohistochemical and RNA-Seq/qRT-PCR data. According to our preliminary results in cell culture systems we observed a transcriptional and translational regulation of NRP2. According to Gemmil et al., NRP2b protein is more stable than NRP2a [37]. Hence, we expected a difference between NRP2 messenger and protein levels. Technical advances and the increasing availability of microdissection, digital spatial profiling or single cell RNA sequencing have made these options more feasible and should be employed in the future.

## 5. Conclusions

In the current study we correlated *NRP*, *PDGFC* and *PDGFD* messenger expression with clinical outcomes in bladder cancer. TCGA data as well as a retrospective single center cohort demonstrated that a high messenger expression of the investigated genes associated with a more aggressive disease (i.e., a high T stage, positive lymph node status and reduced survival). Investigating the relevance of *NRP2* transcripts as prognostic markers, *NRP2A* emerged as an independent prognostic marker for a shorter CSS in bladder cancer patients.

## Figures and Tables

**Figure 1 genes-12-00550-f001:**
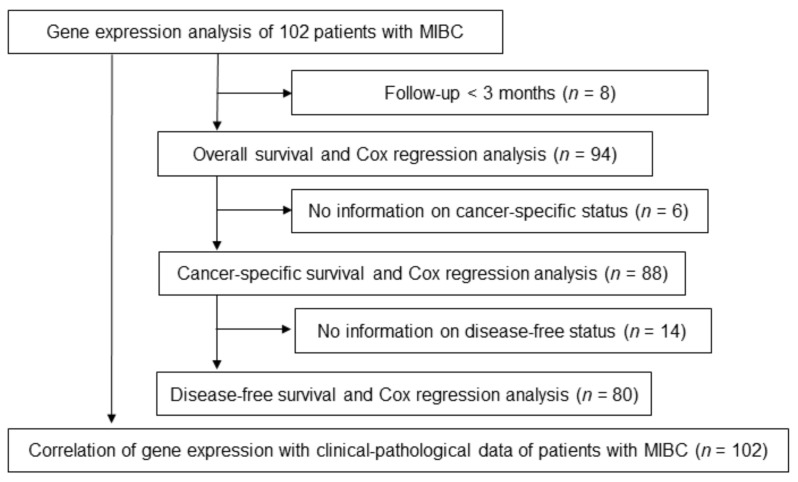
Inclusion and exclusion scheme of the Mannheim cohort. Abbreviations: MIBC, muscle-invasive bladder cancer.

**Figure 2 genes-12-00550-f002:**
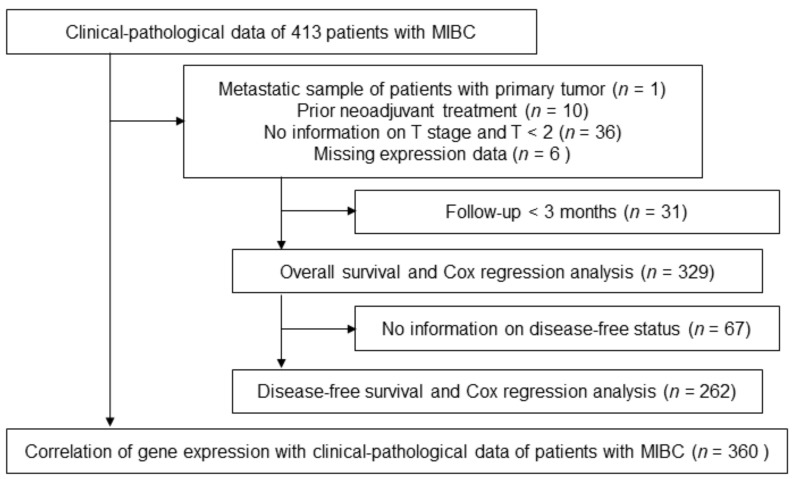
Inclusion and exclusion scheme of The Cancer Genome Atlas (TCGA) cohort. Abbreviations: MIBC, muscle-invasive bladder cancer.

**Figure 3 genes-12-00550-f003:**
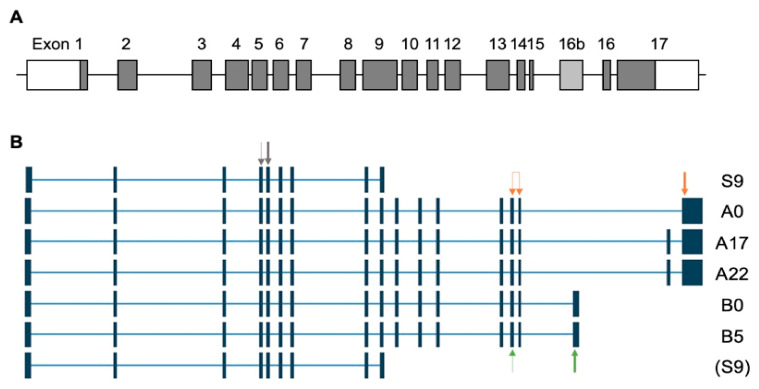
NRP2 gene and NRP2 isoforms/variants observed in the TCGA cohort. (**A**) Genomic organization of the NRP2 gene as identified by Rossignol et al. 2000 [29]. (**B**) *NRP2* splice variants observed as determined by the TCGA Splice Variant database (TSVdb). Splice variants are named according to the resulting *NRP2* transcript. Two different accession numbers for S9 *NRP2* were listed in the TSVdb but only one (top) was expressed in the TCGA BLCA cohort. The primers used for this study are depicted as follows: NRP2: grey arrows; NRP2A: orange arrows (forward primer spans two exons); NRP2B: green arrows. Gene-specific cDNA synthesis was performed with reverse primers (bold arrows), respectively.

**Figure 4 genes-12-00550-f004:**
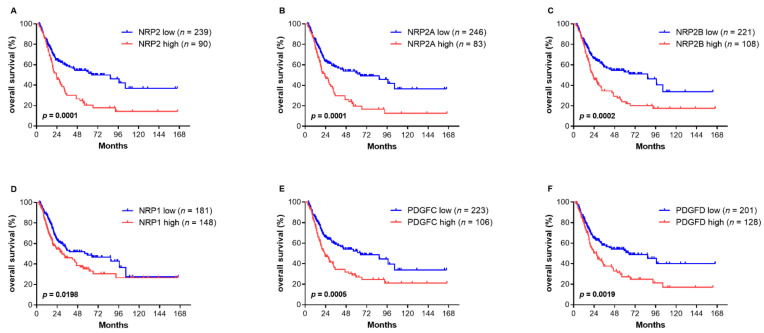
Kaplan–Meier curves of overall survival (OS) stratified by the mRNA expression of NRP2 and its transcripts (**A**–**C**), NRP1 (**D**) and PDGFC (**E**) and PDGFD (**F**). The cut-off values for high and low gene expressions were determined by a partition test with each group representing at least 20% of the total cohort. Significant *p* values are printed in bold.

**Figure 5 genes-12-00550-f005:**
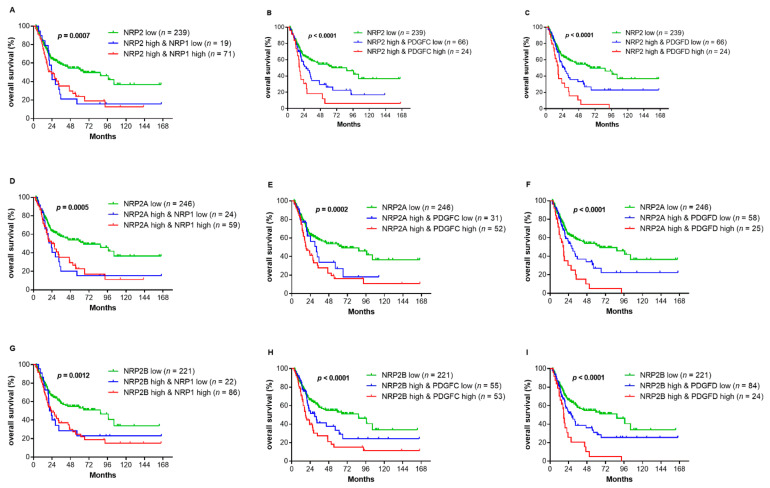
Kaplan–Meier curves of overall survival (OS) of marker combinations. A–C: NRP2 and NRP1 (**A**), PDGFC (**B**) or PDGFD (**C**). D–F: NRP2A and NRP1 (**D**), PDGFC (**E**) or PDGFD (**F**). G–I: NRP2B and NRP1 (**G**), PDGFC (**H**) or PDGFD (**I**). The cut-off values for high and low gene expressions were determined by a partition test with each group representing at least 20% of the total cohort. For the subsequent division of the high expression subgroup, the sample size was adjusted and a partition test with a 20% minimal group size was repeated. Significant *p* values are printed in bold.

**Figure 6 genes-12-00550-f006:**
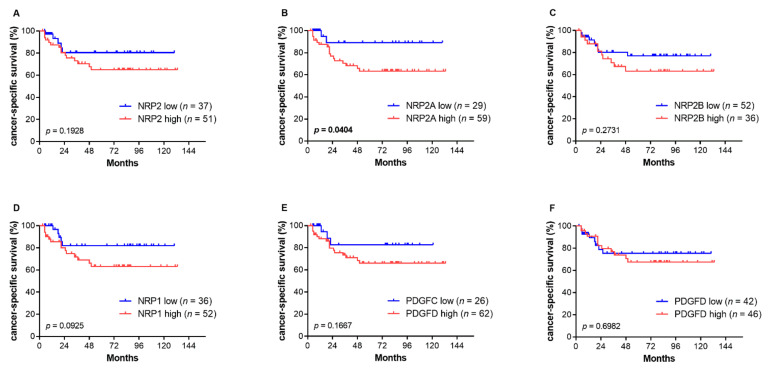
Kaplan–Meier curves of cancer-specific survival (CSS) stratified by mRNA expression of *NRP2* and its transcripts (**A**–**C**), *NRP1* (**D**) and *PDGFC* (**E**) and *PDGFD* (**F**). The cut-off values for high and low gene expressions were determined by a partition test with each group representing at least 20% of the total cohort. Significant *p* values are printed in bold.

**Table 1 genes-12-00550-t001:** Clinicopathological characteristics of the cohorts used in this study.

Characteristics	TCGA Cohort (*n* = 360)		Mannheim Cohort (*n* = 102)	
**Age, y, median (IQR)**	69 (60–77)		71.1 (63.3–77.8)	
**Gender**	N	%	N	%
Male	263	73.1	73	71.6
Female	97	26.9	29	28.4
**Grading (WHO 2017)**				
Low grade	20	5.6	0	0
High grade	338	93.9	100	98.0
Unknown	2	0.6	2	2.0
**Pathological T stage**				
pT1 *	0	0	8 *	7.8
pT2	116	32.2	22	21.6
pT3	188	52.2	53	52.0
pT4	56	16.6	19	18.6
**Pathological N stage**				
pN0	216	60.0	73	71.6
pN1	44	12.2	9	8.8
pN2	70	19.4	11	10.8
pN3	7	1.9	2	2.0
unknown	23	6.4	7	6.9
**LVI**				
No	113	31.4	68	66.7
Yes	141	39.2	34	33.3
Missing	106	29.4	0	0

* Muscle-invasive bladder cancer (MIBC) or Cis were observed in TURBT. Abbreviations: IQR, interquartile range; y, year; LVI, lymphovascular invasion; Cis, carcinoma in situ; TURBT, transurethral resection of the bladder tumor.

**Table 2 genes-12-00550-t002:** Associations of gene expression with clinicopathological parameters in the TCGA cohort.

	NRP1	NRP2	NRP2A	NRP2B	PDGFC	PDGFD
Variable (n)	Median (IQR)	Median (IQR)	Median (IQR)	Median (IQR)	Median (IQR)	Median (IQR)
**Age**						
<70 (186)	9.98 (9.10–10.69)	8.79 (7.33–10.00)	8.75 (7.26–9.86)	5.59 (3.64–6.89)	7.69 (6.51–8.78)	6.78 (5.81–7.71)
≥70 (174)	10.06 (9.04–10.88)	8.99 (7.89–10.01)	8.91 (7.86–10.73)	5.62 (4.19–6.98)	7.81 (6.70–8.95)	6.84 (5.82–7.79)
*p* value	0.4088	0.1810	0.1748	0.3142	0.5045	0.8287
**Gender**						
Female (97))	9.90 (9.04–10.84)	9.14 (7.73–10.12)	9.02 (7.63–10.01)	5.61 (4.00–7.09)	7.87 (6.77–8.79)	6.77 (5.81–7.67)
Male (263)	9.984 (9.10–10.69	8.89 (7.55–9.96)	8.81 (7.51–9.85)	5.61 (3.98–6.91)	7.70 (6.57–8.9)	6.86 (5.81–7.78)
*p* value	0.8083	0.4529	0.4654	0.7337	0.4687	0.8627
**T stage**						
T2 (116)	9.68 (8.65–10.51)	8.20 (6.63–9.53)	8.21 (6.64–9.39)	4.57 (2.84–6.16)	7.29 (6.35–8.32)	6.82 (5.80–7.65)
T3/4 (244)	10.08 (9.28–10.88)	9.26 (8.19–10.12)	9.14 (8.09–9.98)	5.82 (4.49–7.07)	8.00 (6.92–9.04)	6.80 (5.82–7.78)
*p* value	**0.0065**	**<0.0001**	**<0.0001**	**<0.0001**	**0.0002**	0.8132
**N stage**						
N neg (216)	10.03 (9.10–10.75)	8.66 (7.42–9.78)	8.62 (7.39–9.59)	5.39 (3.64–6.82)	7.62 (6.52–8.67)	6.70 (5.86–770)
N pos (121)	9.98 (9.26–10.93)	9.29 (8.23–10.31)	9.18 (8.13–10.10)	5.83 (4.79–7.28)	8.13 (6.94–9.09)	7.13 (5.82–8.06)
*p* value	0.6744	**0.0015**	**0.0012**	**0.0021**	**0.0175**	0.1497
**LVI**						
No (113)	9.85 (8.86–10.85)	8.57 (7.52–9.78)	8.54 (7.48–9.77)	5.28 (3.7–6.86)	7.43 (6.47–6.75)	6.46 (5.56–7.40)
Yes (141)	10.06 (9.34–10.89)	9.15 (7.83–10.01)	9.09 (7.82–9.97)	5.83 (4.46–7.11)	8 (6.75–8.92)	7.18 (6.14–7.88)
*p* value	0.1056	0.1031	0.1226	0.0594	0.0846	**0.0031**
**Recurrence**						
No (159)	9.91 (9.02–10.63)	8.57 (7.32–9.55)	8.49 (7.10–9.48)	5.12 (3.21–6.32)	7.52 (6.38–8.47)	6.67 (5.81–7.47)
Yes (122)	9.97 (9.09–10.89)	9.45 (7.77–10.34)	9.36 (7.71–10.24)	5.92 (4.71–7.56)	7.94 (6.65–9.17)	7.00 (5.74–8.03)
*p* value	0.3203	**0.0003**	**0.0003**	**0.0001**	**0.0364**	0.2689
**Molecular subtypes**						
Basal (132)	10.7 (9.98–11.45)	9.70 (8.90–10.34)	9.58 (8.82–10.21)	6.55 (5.62–7.69)	8.44 (7.45–9.35)	6.00 (5.17–6.93)
Not basal (228)	9.51 (8.85–10.17)	8.36 (7.12–9.43)	8.22 (7.08–9.32)	4.64 (3.04–6.15)	7.41 (6.34–8.39)	7.23 (6.46–8.01)
*p* value	**<0.0001**	**<0.0001**	**<0.0001**	**<0.0001**	**<0.0001**	**<0.0001**

Abbreviations: IQR, interquartile range; LVI, lymphovascular invasion; significant values in bold.

**Table 3 genes-12-00550-t003:** Associations of gene expression with clinicopathological parameters in the Mannheim cohort.

	NRP1	NRP2	NRP2A	NRP2B	PDGFC	PDGFD
Variable (n)	Median (IQR)	Median (IQR)	Median (IQR)	Median (IQR)	Median (IQR)	Median (IQR)
**Age**						
<70 (46)	31.27 (19–32.54)	32.20 (31.15–32.30)	31.72 (31.01–32.18)	19 (19–30.48)	31.74 (19–32.75)	30.39 (19–31.83)
≥70 (54)	31.56 (30.85–32.21)	32.00 (30.72–32.47)	31.33 (30.68–32.06)	28.12 (19–30.30)	31.48 (30.70–32.10)	30.03 (29.01–30.84)
*p* value	0.6464	0.5382	0.2312	0.7446	0.7078	0.8845
**Gender**						
M (73)	31.62 (30.38–32.27)	32.05 (30.75–32.49)	31.60 (30.79–32.08)	26.95 (19–30.42)	31.49 (30.32–32.23)	30.30 (28.70–31.07)
F (29)	31.42 (30.98–32.45)	31.96 (30.93–32.64)	31.60 (30.82–32.11)	28.11 (19–30.39)	31.76 (30.02–32.23)	29.91 (28.62–31.07)
*p* value	0.6687	0.8412	0.9882	0.9622	0.5120	0.7598
**T stage**						
T2 (31)	31.18 (19–32.16)	32.32 (31.21–32.68)	31.68 (30.61–32.14)	29.34 (19–30.61)	31.37 (19–32.06)	30.30 (28.40–31.03)
T3/4 (71)	31.65 (30.83–32.51)	31.80 (30.76–32.47)	31.44 (30.80–32.03)	19 (19–30.29)	31.65 (30.69–32.72)	30.05 (28.77–31.02)
*p* value	0.1355	0.2750	0.9333	0.3526	0.1145	0.9040
**N stage**						
N neg (73)	31.42 (30.38–32.23)	32.04 (30.83–32.49)	31.53 (30.68–32.05)	28.13 (19–30.50)	31.48 (29.78–32.36)	29.91 (28.51–30.88)
N pos (22)	31.58 (29.97–32.93)	31.99 (30.22–32.78)	31.79 (31.01–32.51)	19 (19–29.85)	31.74 (30.74–33.51)	30.64 (26.62–31.82)
*p* value	0.3663	0.7878	0.1554	0.3861	0.4119	*0.0963*
**LVI**						
No (68)	31.18 (30.15–32.14)	31.69 (30.71–32.45)	31.51 (30.65–32.03)	19 (19–30.39)	31.40 (21.67–32.30)	29.89 (21.35–30.78)
Yes (34)	31.98 (31.24–32.64)	32.27 (31.39–32.64)	31.79 (31.17–32.21)	28.57 (19–30.50)	31.91 (31.04–32.99)	30.61 (29.40–31.67)
*p* value	**0.0062**	0.1784	0.0691	0.4297	**0.0653**	**0.0220**
**Recurrence**						
No (48)	31.78 (30.69–32.51)	32.00 (31.19–32.69)	31.591 (30.84–32.09)	28.10 (19–30.45)	31.61 (30.24–32.68)	30.30 (29.17–31.21)
Yes (37)	31.27 (19–32.09)	32.04 (30.26–32.44)	31.70 (30.91–32.11)	26.95 (19–30.53)	31.46 (19–32.37)	30.07 (19–30.79)
*p* value	0.2262	0.32752	0.9188	0.9366	0.5530	0.3675

Abbreviations: IQR, interquartile range; LVI, lymphovascular invasion; significant values in bold.

**Table 4 genes-12-00550-t004:** Univariable and multivariable Cox regression analyses to test the effect of different parameters on overall survival in the TCGA cohort.

	Univariable Analysis			Multivariable Analysis		
Variable	HR	95% CI	*p* value	HR	95% CI	*p* value
**Age** (high/low)	1.55	1.12–2.15	**0.0081**	1.68	1.12–2.56	**0.0130**
**Gender** (male/female)	0.86	0.61–1.24	0.4159			
**T stage** (T3/4/T2)	2.08	1.42–3.15	**0.0001**	1.33	0.80–2.31	0.2838
**N stage** (pos/neg)	2.20	1.57–3.09	**<0.0001**	1.58	0.97–2.58	0.0639
**LVI** (yes/no)	2.12	1.42–3.21	**0.0002**	1.47	0.87–2.49	0.1484
**Molecular subtypes** (basal/not basal)	1.27	0.92–1.76	0.1501			
**NRP1** (high/low)	1.46	1.06–2.03	**0.0208**	1.30	0.81–2.10	0.2809
**NRP2** (high/low)	1.88	1.34–2.60	**0.0003**	1.25	0.37–3.60	0.7023
**NRP2A** (high/low)	1.91	1.36–2.66	**0.0002**	0.84	0.33–2.72	0.7544
**NRP2B** (high/low)	1.82	1.31–2.52	**0.0004**	1.13	0.57–2.17	0.7110
**PDGFC** (high/low)	1.77	1.28–2.44	**0.0007**	1.24	0.73–2.09	0.4284
**PDGFD** (high/low)	1.66	1.20–2.29	**0.0023**	1.61	1.04–2.48	**0.0309**

Abbreviations: HR, hazard ratio; CI, confidence interval; LVI, lymphovascular invasion; significant values in bold.

**Table 5 genes-12-00550-t005:** Summary of Kaplan–Meier analyses for overall survival in the TCGA cohort after stratification for T stage (T2 vs. T3/4) or lymph node metastasis (Nneg vs. Npos).

Variable (n)	NRP1	NRP2	NRP2A	NRP2B	PDGFC	PDGFD
**T stage**						
T2 (107)	**0.0127**	**0.0072**	**0.0003**	**0.0011**	0.0667	**0.0244**
T3/4 (222)	0.1766	**0.0254**	**0.0222**	**0.0114**	**0.0071**	**<0.0001**
**N stage**						
Nneg (197)	**0.0088**	**0.0008**	**0.0007**	**0.0013**	**0.0204**	0.1411
Npos (111)	0.2194	0.2413	0.4104	0.5067	**0.0054**	**<0.0001**

Significant values in bold.

**Table 6 genes-12-00550-t006:** Univariable and multivariable Cox regression analyses to test the effect of different parameters on cancer-specific survival in the Mannheim cohort.

	Univariable Analysis			**Multivariable Analysis**		
Variable	HR	95% CI	*p* value	HR	95% CI	*p* value
**Age** (high/low)	1.40	0.57–3.43	0.4563			
**Gender** (male/female)	1.57	0.63–3.94	0.3485			
**T stage** (T3/4/T2)	2.16	1.80–3.65	**0.0031**	4.00	1.09–14.70	**0.0205**
**N stage** (pos/neg)	2.72	1.03–7.21	0.0614	1.96	0.72–5.31	0.2054
**LVI** (yes/no)	1.95	0.81–4.71	0.1457			
**NRP1** (high/low)	2.32	0.84–6.38	0.0845	1.02	0.32–3.31	0.9690
**NRP2** (high/low)	1.93	0.70–5.32	0.1816			
**NRP2A** (high/low)	4.07	0.94–17.54	**0.0242**	4.50	0.86–23.57	**0.0498**
**NRP2B** (high/low)	1.63	0.67–3.93	0.2786			
**PDGFC** (high/low)	2.31	0.68–7.89	0.1425			
**PDGFD** (high/low)	1.19	0.49–2.92	0.6985			

Abbreviations: HR, hazard ratio; CI, confidence interval; LVI, lymphovascular invasion; significant values in bold.

**Table 7 genes-12-00550-t007:** Summary of Kaplan–Meier analyses for cancer-specific survival in the Mannheim cohort after stratification for T stage (T2 * vs. T3/4) or lymph node metastasis (Nneg vs. Npos *).

Variable (n)	NRP1	NRP2	NRP2A	NRP2B	PDGFC	PDGFD
**T stage**						
T3/4 (60)	0.1985	0.2347	**0.0394**	0.2201	0.1518	0.5592
**N stage**						
Nneg (67)	0.1346	0.2244	0.0651	0.1938	0.1004	0.2428

* Due to the low number of T2 and N positive cases, the statistical power of the stratification was not possible; significant values in bold.

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
