# Peer review of "Neuropilin-2 and Its Transcript Variants Correlate with Clinical Outcome in Bladder Cancer"

_genes, 2021, doi:10.3390/genes12040550_

Round 1

Reviewer 1 Report

The major concern is the experimental design. The authors should have analyzed their own data first and then looked for validation in independent series of cases, preferably using the same methodology instead of looking first on TCGA data. I strongly recomment to use two independent sets of cases with their own methodology and similar number of cases and additionally if needed comparison with TCGA data.

-The different ratio of patients and number of cases with very should follow-up always has a strong weight on outcome analyses.

-Another issue of relevance is the different methodology used in the different cohorts that poses concerns on data interpretation.

  • The criteria for selection of cutoffs for their outcome analyses should be further justified with consensus among sets.
  • Have the authors checked on criteria among series and hospitals for measuring number of positive lymph nodes among series? have they found differences regarding the number of positive nodes with relevance?
  • Kaplan Mayer figures without statistcal significance should not be displayed in the text, maybe as supplemental data.

Author Response

We thank the editors and reviewers for their valuable time in extensively reviewing our manuscript and providing us with insightful comments. We believe the comments and suggestions significantly improved our manuscript.

After careful consideration of each individual comment, we have implemented significant changes in the revised version of our manuscript to address the reviewers’ and editors’ concerns. Following are the responses to the comments:

“The major concern is the experimental design. The authors should have analyzed their own data first and then looked for validation in independent series of cases, preferably using the same methodology instead of looking first on TCGA data. “

We are grateful for this very valuable suggestion. In the TCGA database only overall survival and disease free survival was recorded. To add additional important data we used the Mannheim cohort. The Mannheim cohort had the big advantage of having access to cancer specific survival data. Therefore, our story line goes from overall survival to the more specific disease specific survival. Nonetheless, the issue of methodology, sample size etc. was addressed in methods, results and discussion sections as described below.

“I strongly recommend to use two independent sets of cases with their own methodology and similar number of cases and additionally if needed comparison with TCGA data. “

Again, the authors agree. Unfortunately due to the time restraints for the resubmission (10 days) it is not possible to use a validation cohort because we do not have any other cohort at hand at the moment. Furthermore, we discussed differences between TCGA data and data generated from the Mannheim cohort:

„These discrepancies [regarding NRP2 transcripts and survival] may arise due to different sample collection and preparation (e.g. fresh frozen vs. FFPE material and RNASeq vs. qRT-PCR) in the TCGA and Mannheim cohorts. Investigations using another independent cohort could clarify these issues but unfortunately, we do not have access to another dataset providing information on NRP2 transcripts at the moment.”

“The different ratio of patients and number of cases with very should follow-up always has a strong weight on outcome analyses.”

We agree with the Reviewer that differences in sample sets may have an effect on outcome but unfortunately it is not always possible to find matching sample sets. Nonetheless, to address this issue the following changes were made:

In results (section 3.1) the sentence was added “While demographic and clinopathological distribution of the Mannheim cohort (Table 1) was similar to the TCGA dataset, the TCGA cohort was larger than the Mannheim cohort (360 vs. 102 patients) but the Mannheim cohort had significantly longer follow up times (TCGA: 29.8 months versus Mannheim 82 months).“

In the discussion section “follow-up time” was added to list in the sentence “Before discussion of the results obtained in our current study, it should be pointed out that the TCGA cohort analyzed fresh frozen material while the Mannheim cohort consisted of FFPE material. While fixation is known to affect the sample, FFPE material has the great advantage of being routinely collected and stored in the clinic usually making it more easily accessible.” Additionally we added: “These discrepancies may arise due to differences in study cohorts (sample acquisition by RC vs. TURBT; treatment-naïve patients vs. radiochemotherapy; sample size; sample type (fresh frozen vs. FFPE); follow-up time).”

-Another issue of relevance is the different methodology used in the different cohorts that poses concerns on data interpretation.

To address this issue, the following sentences were added

Material and Methods, section TCGA BLCA cohort: “mRNA was extracted from fresh frozen samples, prepared into libraries and sequenced by Illumina HiSeq as described in (4).”

Discussion: “Before discussion of the results obtained in our current study, it should be pointed out that the TCGA cohort analyzed fresh frozen material while the Mannheim cohort consisted of FFPE material.”

  • The criteria for selection of cutoffs for their outcome analyses should be further justified with consensus among sets.

To clarify that both cohorts were analyzed using the same statistical methods, Method section “2.3 TCGA cohort and statistics” was separated into “TCGA BLCA cohort” and “statistics” and the following sentence was added: “Analyses of both cohorts were performed using the methods described below.”

Cut-off values were determined as stated in the Material and Methods section and as previously published [27, 28] and where appropriate, the following sentence was included into the figure legends: “The cut-off values for high and low gene expression were determined by partition test, with each group representing at least 20 % of the total cohort.”

For marker combinations (Figure 5), the following sentences were added to the figure legend: “The cut-off values for high and low gene expression were determined by partition test, with each group representing at least 20 % of the total cohort. For subsequent division of the high expression subgroup, sample size was adjusted and partition test with 20% minimal group size was repeated.”

  • Have the authors checked on criteria among series and hospitals for measuring number of positive lymph nodes among series? have they found differences regarding the number of positive nodes with relevance?

If we include all this data into our analysis, the sample size would be too small to make statistically valid statements.

  • Kaplan Mayer figures without statistical significance should not be displayed in the text, maybe as supplemental data.

We understand the Reviewers point but for consistency reasons and to simplify comparison between cohorts and figures, we decided not to remove Kaplan Meier plots without statistical significance.

Reviewer 2 Report

The manuscript is well written and presents results on an interesting set of genes. I have no critique on the language or text, but I find that some clarification of the roles of the investigated genes is warranted. As stated by the authors, the main shortcoming of the work is the sole reliance on qRT-PCR analysis which somewhat reduces the novelty of the work.

Points to address:

The most pressing issue the authors should address is the source of the measured expression. The Neuropilin and PDGF expression (with the exception of PDGFD) very closely follows general estimation of stromal content for tumors in the TCGA and other dataset (e.g. through ECM genes ACTA2, VIM, ZEB2 etc, with correlations ranging between 0.7 – 0.9). This indicates that the measured expression levels are at least partly determined by the overall amount of stromal other and non-urothelial cells in the biopsy. The amount of non-urothelial cells in a biopsy may be linked to intrinsic invasive/infiltrative properties of the tumor, but a bulk biopsy from a higher stage tumor is almost by definition more likely to include cells from the surrounding invaded tissue, thus correlating with stage and in turn with other clinical risk factors. If expression primarily occurs in the non-tumor cells, the measured expression levels could potentially also be impacted by actual physical sampling of the tumor. As the investigators utilize mRNA expression where the source cannot be determined it would be beneficial to provide a more expanded description of the expected expression source and the biological functions of the study genes/isoforms in tumor vs non-tumor cells in the introduction to give more context to the manuscript.

Suggestions:

A comparison to general stromal signatures (e.g. ESTIMATE) in the TCGA dataset could be performed to give an indication of whether the study genes provide additional information.

The associations within the TCGA dataset could be examined further in the context of the molecular subtypes to evaluate their value across tumors with shared underlying biology. Tumors with Luminal/Urothelial-like/LumP,LumNS,LumU are overall less infiltrated (immune and stoma) than subtypes with known higher clinical risks like Basal/BaSq-like/BaSq with the MDA/Lund/Consensus nomenclature, or corresponding TCGA subtypes.

Author Response

We thank the editors and reviewers for their valuable time in extensively reviewing our manuscript and providing us with insightful comments. We believe the comments and suggestions significantly improved our manuscript. 

After careful consideration of each individual comment, we have implemented significant changes in the revised version of our manuscript to address the reviewers’ and editors’ concerns. Following are the responses to the comments:

The manuscript is well written and presents results on an interesting set of genes. I have no critique on the language or text, but I find that some clarification of the roles of the investigated genes is warranted. As stated by the authors, the main shortcoming of the work is the sole reliance on qRT-PCR analysis which somewhat reduces the novelty of the work.

Point by point response:

The most pressing issue the authors should address is the source of the measured expression. The Neuropilin and PDGF expression (with the exception of PDGFD) very closely follows general estimation of stromal content for tumors in the TCGA and other dataset (e.g. through ECM genes ACTA2, VIM, ZEB2 etc, with correlations ranging between 0.7 – 0.9). This indicates that the measured expression levels are at least partly determined by the overall amount of stromal other and non-urothelial cells in the biopsy. The amount of non-urothelial cells in a biopsy may be linked to intrinsic invasive/infiltrative properties of the tumor, but a bulk biopsy from a higher stage tumor is almost by definition more likely to include cells from the surrounding invaded tissue, thus correlating with stage and in turn with other clinical risk factors. If expression primarily occurs in the non-tumor cells, the measured expression levels could potentially also be impacted by actual physical sampling of the tumor. As the investigators utilize mRNA expression where the source cannot be determined it would be beneficial to provide a more expanded description of the expected expression source and the biological functions of the study genes/isoforms in tumor vs non-tumor cells in the introduction to give more context to the manuscript.

We are thankful to the reviewer for pointing out this very important issue. To see where our proteins of interest are expressed we accessed the human protein atlas and evaluated the immunohistochemistry. For NRP1 and NRP2 we detected most of the staining in the tumor cells and only some staining in the surrounding tissue. Without doubt smooth muscle cells express NRP2 and this would be a caveat of this study. For PDGFC and PDGFD we also detected some expression in the surrounding tissue which might vary from case to case.

Hence, several sentences/paragraphs were added addressing this issue:

Introduction: “Like NRPs, PDGFs and PDGF receptors are abundantly expressed by cells in the tumor microenvironment (e.g. endothelial cells, vascular smooth muscle cells, and tumor-associated macrophages) but also by tumor cells”

Discussion (after “Intriguingly, the prognostic value of PDGFs was dependent on tumor versus stromal expression of the analyzed proteins, i.e. stromal PDGFD could predict CSS, in contrast, tumor but not stromal PDGFC could predict CSS [36].”): “The human protein atlas (HPA) provides some information on the protein expression of NRP2 in bladder cancer tissue. According to the date in the HPA a subgroup of cancer tissue shows no expression of NRP2 protein while most of the urothelial cancer tissue express high levels of NRP2. NRP1 can be also detected in high amounts in urothelial cancer cells. In general, due to stromal and immune cell infiltration, tumor samples contain a mixture of cells with potentially distinct expression levels of specific genes. Determining tumor purity can provide important additional information. Hence, pathologists routinely determine tumor purity based on hematoxylin- and eosin-stained slides and combine this information with immunohistochemical stainings. According to the HPA most of the staining was localized in the tumor cells. In addition, tumor purity can be investigated using ABSOLUTE or ESTIMATE methods based on genomic or transcriptomic information [49, 50]. Indeed, in the TCGA cohort of muscle invasive tumors, tumor purity (based on ABSOLUTE) significantly decreased while stromal score and ESTIMATE score significantly increased in more aggressive disease (data not shown).“

 Suggestions:

A comparison to general stromal signatures (e.g. ESTIMATE) in the TCGA dataset could be performed to give an indication of whether the study genes provide additional information.

We greatly appreciate the Reviewer´s comment as we were not aware of ESTIMATE. Stromal, Immune and ESTIMATE scores were investigated and in fact, high stromal scores were associated with higher T stage, N positive stage and LVI while immune score were not. The resulting ESTIMATE score was still associated with higher T stage. Also, with the exception of PDGFD, gene expression correlated with ESTIMATE score. As to not excessively alter and lengthen the manuscript, these analyses were not included in the Material and Methods and Results sections. Instead, the following paragraph was added to the discussion (see also comment above):

“The human protein atlas (HPA) provides some information on the protein expression of NRP2 in bladder cancer tissue. According to the date in the HPA a subgroup of cancer tissue shows no expression of NRP2 protein while most of the urothelial cancer tissue express high levels of NRP2. NRP1 can be also detected in high amounts in urothelial cancer cells. In general, due to stromal and immune cell infiltration, tumor samples contain a mixture of cells with potentially distinct expression levels of specific genes. Determining tumor purity can provide important additional information. Hence, pathologists routinely determine tumor purity based on hematoxylin- and eosin-stained slides and combine this information with immunohistochemical stainings. According to the HPA most of the staining was localized in the tumor cells. In addition, tumor purity can be investigated using ABSOLUTE or ESTIMATE methods based on genomic or transcriptomic information [49, 50]. Indeed, in the TCGA cohort of muscle invasive tumors, tumor purity (based on ABSOLUTE) significantly decreased while stromal score and ESTIMATE score significantly increased in more aggressive disease (data not shown). “

The associations within the TCGA dataset could be examined further in the context of the molecular subtypes to evaluate their value across tumors with shared underlying biology. Tumors with Luminal/Urothelial-like/LumP,LumNS,LumU are overall less infiltrated (immune and stoma) than subtypes with known higher clinical risks like Basal/BaSq-like/BaSq with the MDA/Lund/Consensus nomenclature, or corresponding TCGA subtypes.

The Reviewer raises a valid point. As described by Robertson et al (PMID: 28988769), muscle-invasive bladder cancer can be divided into five molecular subtypes (basal-squamous, luminal-papillary, luminal-infiltrated, luminal, and neuronal); for statistical reasons, we dichotomized these subtypes into “basal” and “not basal” and included the results in our study (results included in tables 3, 5).

The following paragraphs, sentences were added to the manuscript:

Introduction: MIBC is a heterogeneous disease that shows high overall mutation rates. Based on transcriptome profiling/mRNA expression, MIBC can be divided into five molecular subtypes:  basal-squamous, luminal-papillary, luminal-infiltrated, luminal, and neuronal subtypes (Robertson et al 2017 PMID: 28988769). These molecular subtypes are of clinical significance as they may be used to stratify patients for prognosis and response to chemo- or immunotherapy (Kamoun et al 2020; Robertson et al 2017).

Results (3.4. Associations between marker expression and clinicopathological characteristics): Lastly, association of gene expression with molecular subtype was investigated. Based on molecular signatures, TCGA samples were grouped into basal_squamous (n = 132), luminal, luminal_infiltrated, luminal_papillary and neuronal (n = 25, 69, 118, 16, respectively). Molecular subtype complexity was reduced by dichotomizing into basal versus not basal type. For all investigated genes, gene expression significantly associated with molecular subtype in the TCGA cohort. With the exception of PDGFD, higher gene expression was always observed in basal subtype.

Reviewer 3 Report

In their paper, Förster and colleagues explored the expression of Neuropilin-1 and 2 (NRP1 and 2), NRP2 isoforms and ligands PDGFC and PDGFD in muscle-invasive bladder cancer in relation to overall, disease-free and cancer-specific survival. This is an extension of their previous work published in 2015 (PMID: 24862180) but using other ligands not previously analyzed (VEGF-C before and PDGFC and PDGFD now). Some justification for the choice of ligands used in various analyses would be helpful.

Figure 3 could show the location of primers for cDNA synthesis.

TCGA expression is based on fresh-frozen tissues, while the Mannheim cohort – on FFPE tissues. This should be specifically discussed, pointing out to possible benefits/limitations of this approach.

In TCGA - patients with missing gene expression data were excluded. What is considered to be a missing expression? What about tumors with non-detectable expression? What was the expression cutoff for inclusion?

NRP2 positively correlated with NRP1, PDGFC, and PDGFD – Why would NRP2 transcripts correlate with NRP1 and the ligands? Any thoughts?

On all survival plots – please indicate the sample numbers in each comparison group. It is unclear how these groups are defined. Is it based on below/above median expression levels?

The Human Protein Atlas shows the expression of these genes both on mRNA and protein levels and this should be checked and discussed.

It is unclear how the TCGA dataset was reanalyzed, no analysis details are provided.

Author Response

We thank the editors and reviewers for their valuable time in extensively reviewing our manuscript and providing us with insightful comments. We believe the comments and suggestions significantly improved our manuscript.

After careful consideration of each individual comment, we have implemented significant changes in the revised version of our manuscript to address the reviewers’ and editors’ concerns. Following are the responses to the comments:

In their paper, Förster and colleagues explored the expression of Neuropilin-1 and 2 (NRP1 and 2), NRP2 isoforms and ligands PDGFC and PDGFD in muscle-invasive bladder cancer in relation to overall, disease-free and cancer-specific survival. This is an extension of their previous work published in 2015 (PMID: 24862180) but using other ligands not previously analyzed (VEGF-C before and PDGFC and PDGFD now).  Some justification for the choice of ligands used in various analyses would be helpful.

To explain the choice of ligands, the following paragraph was added to the introduction:

“In addition to semaphorins and VEGFs, also other heparin-binding growth factors like platelet-derived growth factors (PDGFs) have been described to bind to NRPs [13, 14] and several reports have indicated involvement of NRPs in PDGF signaling [15-17]. The PDGF family consists of four members (PDGF-A through D) that are secreted as homo- or heterodimers (AA, AB, BB, CC or DD) and bind to and signal via PDGF receptors. Like NRPs, PDGFs and PDGF receptors are abundantly expressed by cells in the tumor microenvironment (e.g. endothelial cells, vascular smooth muscle cells, and tumor-associated macrophages) but also by tumor cells (reviewed in [18]). Due to their more recent discovery, the role of PDGF-C and PDGF-D in cancer is less well studied.”

Figure 3 could show the location of primers for cDNA synthesis.

We added the location of the primers in Figure 3.

TCGA expression is based on fresh-frozen tissues, while the Mannheim cohort – on FFPE tissues. This should be specifically discussed, pointing out to possible benefits/limitations of this approach.

As the reviewer correctly points out, TCGA and the Mannheim cohort used differently fixed/stored tissue. For quality reasons as well as for better comparison, using fresh/fresh frozen material would have been better. However, access to FF material is usually limited while FFPE samples are more easily accessible. To accommodate the lower quality or partial degradation of FFPE samples, we specifically designed primers to amplify relatively short fragments. This information is now included in the Methods section:

“Primers were specifically designed to generate short (approx. 150 bp) amplification products from FFPE material.”

To point out that TCGA and Mannheim cohort used different samples, the following sentence was added to the discussion:

“Before discussion of the results obtained in our current study, it should be pointed out that the TCGA cohort analyzed fresh frozen material while the Mannheim cohort consisted of FFPE material. While fixation is known to affect the sample, FFPE material has the great advantage of being routinely collected and stored in the clinic making it more easily accessible.”

In TCGA - patients with missing gene expression data were excluded. What is considered to be a missing expression? What about tumors with non-detectable expression? What was the expression cutoff for inclusion?

Upon downloading the expression data from the TCGA cohort via Xenabrowser, six patients had no data for the investigated genes, i.e. the cell in the downloaded data table was empty. These samples were therefore excluded from gene expression analysis. In all other samples, log2 transformed expression of the investigated genes was detectable at varying levels (4.3 to 12) and all samples were included in the analysis.

NRP2 positively correlated with NRP1, PDGFC, and PDGFD – Why would NRP2 transcripts correlate with NRP1 and the ligands? Any thoughts?

It is known that NRP1 and NRP2 form heterodimers which might explain the correlation of NRP1 and NRP2 expression and also the correlation of NRP1 ligands with NRP2 expression. The functional relevance of the NRP1/NRP2 heterodimers is still under investigation.

We added the following paragraph in the discussion:

“NRPs can form homo- and heterodimers and interact with similar ligands. While our data shows a positive correlation between NRP2 and NRP1 expression, in vitro results regarding NRP interaction/regulation are inconsistent [38, 39].”

Thoughts on NRPs & PDGFs:

In agreement with our results, correlation of NRPs with PDGFs is also observed in cell culture studies and the following paragraph was added to the discussion:

“Several cell culture studies indicated NRPs in PDGF signaling. In smooth muscle cells, PDGF-BB upregulates NRP1 mRNA levels and vice versa, NRP1 as well as NRP2 knockdown reduced PDGF-AA and/or PDGF-BB-induced PDGF receptor phophorylation [15-17]. Here, we also observe a positive correlation between NRP2 and its transcripts and PDGFC (and in the Mannheim cohort also PDGFD). All transmembrane NRP2 proteins have the same extracellular ligand binding domains and could thus interact with the same ligands. To date, it is unclear how PDGF and NRP expression are linked mechanistically.”

On all survival plots – please indicate the sample numbers in each comparison group. It is unclear how these groups are defined. Is it based on below/above median expression levels?

Sample numbers were added to the figures as requested. The groups were determined by partition test using Jmp software as previously described [27, 28]. This is also described in the material and methods section:  “The cut-off values for high and low gene expression were determined by partition test, with each group representing at least 20 % of the total cohort.” Nonetheless, this information was also included in the figure legends.

The Human Protein Atlas shows the expression of these genes both on mRNA and protein levels and this should be checked and discussed.

We added information about NRP2 protein and messenger expression in the discussion section of our manuscript. According to the Human Protein Atlas, most of the urothelial cancer tissue shows high protein expression of NRP2. The stromal tissue shows only a few positive cells. As expected NRP2 messenger levels correlate with outcome in bladder cancer. It is important to emphasize that our experimental data hint to different levels of messenger and protein due to transcriptional and translational control mechanisms that we found in our cell culture experiments.

It is unclear how the TCGA dataset was reanalyzed, no analysis details are provided.

TCGA data and the Mannheim cohort were analyzed using the same methods as described in the material and method section. To clarify this aspect, the paragraph “2.3 TCGA cohort and statistics” in the methods section was separated into “TCGA BLCA cohort” and “statistics” and the following sentence was added after “Finally, the expression and clinical data of 360 patients with MIBC were reanalyzed.”:

“Correlation analyses between different genes, survival analyses and Cox regression analysis were performed as described in 2.4. Statistics”.